# MoMamba: A Lightweight Music Oriented Mamba-Based Model for Music Information Retrieval Tasks

## Abstract

Music Information Retrieval (MIR) tasks on raw audio have traditionally been tackled using convolutional neural networks (CNNs) and transformer-based models. While CNNs effectively capture local structures and transformers leverage attention for long-range dependencies, both architectures come with computational and scalability challenges. In this study, we introduce a novel extension of Mamba tailored to music. Our resulting method, MoMamba (Music Oriented Mamba), is a lightweight Mamba-based music classification model. We evaluate MoMamba's performance across several benchmark MIR tasks. Our results show that MoMamba consistently outperforms a number of baselines, including an existing Mamba-based method, on all of the benchmark datasets we considered. Importantly, all models were trained from scratch without any pretraining, making the performance gains especially notable since they cannot be attributed to transfer learning. Additionally, our model's performance rivals existing benchmarks from models pretrained on much larger datasets. Our work highlights the advantages of MoMamba in music analysis and retrieval such as accuracy and inference time, encouraging further research into its capabilities within the MIR domain.

## 1 Introduction

Extracting features from music is essential for understanding its structure, patterns, and meaning. These features enable various applications, from classification and recommendation systems to composition and analysis (Copet et al. (2023); Dhariwal et al. (2020); Schedl et al. (2018); Hernandez-Olivan & Beltran (2021)). By identifying musical characteristics, we can enhance both human and machine interpretation, thereby improving organization, retrieval, and interaction with music.

Traditionally, deep learning approaches for music downstream classification tasks have relied heavily on Convolutional Neural Networks (CNNs) (Elbir & Aydin (2020); Bian et al. (2019)). CNNs are particularly effective in processing spectrogram representations of audio but struggle to capture global context effectively. Convolutional Recurrent Neural Networks (CRNNs) (Choi et al. (2017)) combine CNNs for feature extraction with Recurrent Neural Networks (RNNs) for sequential modeling. While CRNNs have demonstrated strong performance in MIR tasks (Choi et al. (2017)), the design of the model inherently has challenges such as vanishing/exploding gradients and global locality problems. More recently, attention-based models have emerged, leading to a shift toward Audio Spectrogram Transformers (ASTs) (Gong et al. (2021)). They leverage self-attention mechanisms to model long-range dependencies and capture global context. However, they come with high computational cost, require substantial pretraining data, and have quadratic complexity.

Recently, SSMs have gained attention in natural language processing (NLP) due to their ability to model long-range dependencies while maintaining computational efficiency (Gu & Dao (2024)). Their success in NLP has sparked interest in their applicability to other sequential domains, including audio and music processing. While the use of SSMs in audio is still in its early stages, recent efforts have demonstrated their potential in various tasks. Notable work such as AudioMamba (Erol et al. (2024)) has showcased promising results in classifying speech and sound effects, illustrating SSMs' ability to effectively model temporal structures in audio signals. These advancements suggest that SSMs provide a viable alternative to traditional deep learning architectures, particularly

for music-related applications where capturing both local and long-range dependencies is essential. Our method also outperforms ASTs on average in downstream tasks in terms of performance and inference speed but we make significant changes compared to AudioMamba, which simply replaces the transformer layers in ASTs with Mamba layers: we utilize the entire spectrogram without patchification in order to keep the spectrogram as one intact sequence and remove the positional encoding.

We note that pretrained foundational models have demonstrated strong performance on the MIR tasks we consider (Li et al. (2024); Castellon et al. (2021); McCallum et al. (2022)). However, these methods rely on large-scale external datasets, which makes it difficult to separate the effects of the architecture from the benefits of pretraining. For this reason, our main comparisons are with non-pretrained models, but we also provide the comparison to MERT (Li et al. (2024)) and pretrained models from (McCallum et al. (2022)). Once large pretrained Mamba models become available, a more direct comparison with other pretrained approaches will be possible.

We evaluate multiple model setups to determine the effects of each component. Based on these experiments, our contributions include developing MoMamba, a Mamba-based music classification model with a class token, and demonstrating its strong performance across multiple downstream music tasks.

Our contributions:

- **We develop a Mamba-based music classification model with a class token mechanism (MoMamba):** We leverage the Mamba architecture to process **entire** audio spectrograms, and introduce a class token at the **end** of the sequence to enhance classification performance. To our knowledge, we are the first to apply Mamba to these MIR downstream tasks.

- **We demonstrate that MoMamba achieves competitive performance on downstream tasks:** Through extensive experiments with training from scratch, MoMamba matches or surpasses the performance of existing models on seven music classification benchmarks, while also offering superior inference time when compared to ASTs and RNNs. MoMamba also outperforms AudioMamba in every metric, and performs comparably to pretrained foundational models that utilize orders of magnitude more training data.

## 2 RELATED WORK

Early deep learning approaches to raw audio music classification are primarily CNN-based, leveraging architectures such as ResNet18 (Elbir & Aydin (2020)) and DenseNet (Bian et al. (2019)). These models typically process spectrograms that have been compressed into smaller images (e.g., $128 \times 128$) before being fed into the network. In meter detection, ResNet18 demonstrates superior performance to both traditional non-machine-learning techniques and supervised non-deep-learning models (Abimbola et al. (2024)).

To address the sequential nature of music data, CRNNs serve as a hybrid approach (Choi et al. (2017)), combining CNNs for feature extraction with RNNs for temporal modeling. This architecture allows for more effective processing of spectrograms by capturing both spatial and sequential patterns, making it particularly well-suited for MIR downstream tasks. By leveraging RNN components such as LSTMs or GRUs, CRNNs improve upon pure CNN models by incorporating temporal dependencies.

ASTs have emerged as a more effective alternative for audio-related tasks thanks to their attention mechanism. Unlike CNNs, which rely on localized feature extraction, transformers process entire sequences, enabling them to capture long-range dependencies and improve overall performance (Gong et al. (2021)).

Transformer-based models have driven major advances in music generation and analysis, with notable impact on downstream MIR tasks. For example, MusicGen (Copet et al. (2023)) is an autoregressive transformer that uses EnCodec (Défossez et al. (2022)) for audio tokenization, and its internal representations have been shown to transfer effectively to tasks such as instrument recognition and music tagging (Koo et al. (2024)). JukeBox (Dhariwal et al. (2020)) takes a different approach, employing a sparse transformer over multi-scale VQ-VAE codes to generate raw audio with singing

and achieving coherence over several minutes. Beyond generation, the learned representations from JukeBox have also proven useful for MIR tasks like genre classification and emotion recognition (Castellon et al. (2021)). Together, these works illustrate the versatility of transformer-based models in capturing rich musical structure and providing transferable features for analysis.

However, training such models requires extensive datasets. For instance, MusicGen was trained on 20,000 hours of licensed music (Copet et al. (2023)), and JukeBox utilizes a dataset of 1.2 million songs (Dhariwal et al. (2020)). This poses challenges for MIR tasks with limited labeled data, as small datasets can lead to overfitting and hinder the models' ability to generalize effectively. To mitigate this, techniques such as data augmentation, transfer learning, and the use of synthetic data have been explored.

More recently, pretrained models such as MERT (Li et al. (2024)) have been introduced as foundation models for music understanding. MERT is trained on 160 thousand hours of music audio and leverages self-supervised pretraining to provide general-purpose embeddings that have proven highly effective across a range of MIR benchmarks. These models represent an exciting direction, but their reliance on massive pretraining corpora raises questions about accessibility and applicability in domains where such resources are unavailable.

AudioMamba (Erol et al. (2024)), a Mamba-based spectrogram classifier, performs well in speech and sound effects (SFX) classification tasks. Furthermore, AudioMamba shows that Mamba-based models are more memory efficient than transformer-based models. This success suggests that Mamba-based architectures could offer advantages in sequential audio modeling. However, unlike speech or short sound effects, music often contains long-range harmonic and rhythmic structures. Since the general design of AudioMamba is very similar to ASTs, it ignores a main benefit of Mamba-based models: the ability to process long spectrograms sequentially. We address this problem with the design of MoMamba. In this study, we explore Mamba-based architectures for MIR tasks, evaluating their effectiveness across classification benchmarks to assess their viability against other architectures.

## 3 MODEL APPROACH

State Space Models (SSMs) (Gu et al. (2021)) are mathematical models designed for sequence-to-sequence transformations. In general, SSMs, including Mamba, map an input signal $u \in \mathbb{R}^D$ to a hidden state $h \in \mathbb{R}^H$, and then to an output signal $y \in \mathbb{R}^D$, where $H$ is the hidden state size and $D$ is the number of channels.

### 3.1 PRELIMINARIES

There are four continuous parameters used in the Mamba model (Gu & Dao (2024)): $\Delta \in \mathbb{R}, A \in \mathbb{R}^{H \times H}, B \in \mathbb{R}^{H \times D}, C \in \mathbb{R}^{D \times H}$, where we use the frequency bins as channels. $\Delta$ controls the discretization resolution, $A$ is the state transition matrix, $B$ maps the input sequence to the hidden state, and $C$ maps the hidden state to the output state. Specifically, $\Delta$ changes the focus on the current timestep $u_t$: larger values emphasize $u_t$, while smaller values allow the previous state to dominate. In Mamba, the parameters $\Delta, B$, and $C$ are dependent on the input signal. To apply them effectively, the model must first be discretized, converting its continuous-time parameters into discrete counterparts by using fixed formulas $f_a(\Delta, A)$ and $f_b(\Delta, A, B)$. Mamba's discretization (Gu & Dao (2024)) is formulated as:

$$
\begin{aligned}
\overline{A} &= f_a(\Delta, A) = \exp(\Delta A) \\
\overline{B} &= f_b(\Delta, A, B) = (\Delta A)^{-1}(\overline{A} - I) \cdot (\Delta B).
\end{aligned}
\tag{1}
$$

$C$ does not need to be discretized since it is just a linear projection from the hidden state to the output. Finally, we have four learnable parameters: $\Delta, \overline{A}, \overline{B}, C$.

Once discretized, these parameters allow for efficient sequence modeling. When using spectrograms with state space models, each frequency bin is processed independently using the same model parameters during the forward pass. Let $u^{(d)} \in \mathbb{R}^L$ be a sequence for the $d$-th frequency bin, where $L$ represents the number of timesteps in the spectrogram. The sequence transformation can be efficiently computed with the kernel $\overline{K}$ defined below. Using the discretized parameters, the SSM

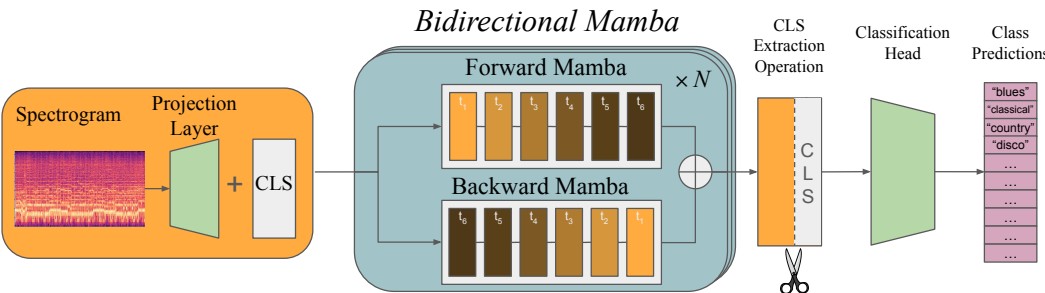

Figure 1: An overview of MoMamba. First, the spectrogram is projected to a higher dimension, followed by the concatenation of the class token. Then, these features pass through the Mamba layers. Finally, the class token is extracted and fed through the classification head. The final output of the model is the class probability distribution.

updates its hidden state and computes the output as follows:

$$
\begin{aligned}
h_t^{(d)} &= \overline{A} h_{t-1}^{(d)} + \overline{B} u_t^{(d)} \\
y_t^{(d)} &= C h_t^{(d)} \\
\overline{K} &= (C\overline{B}, C\overline{AB}, \dots, C\overline{A}^k\overline{B}, \dots) \\
y^{(d)} &= \overline{K} \cdot u^{(d)}.
\end{aligned}
\tag{2}
$$

The output $y^{(d)} \in \mathbb{R}^L$ learns information throughout the whole sequence. For more details, refer to Gu & Dao (2024).

## 3.2 MoMamba Architecture

MoMamba first projects the spectrogram generated from the audio sample into a higher-dimensional space, then appends a class token. The resulting features pass through $N$ bidirectional Mamba layers. Afterward, the class token is extracted by selecting the final timestep of the latent representation. Finally, the resulting vector is projected to the class dimension, and applying softmax produces a probability distribution.

One of the main differences between MoMamba and AudioMamba is the length of each sample. Since AudioMamba is evaluated on speech and sound effects, samples in their work are 10 seconds long, resulting in a shorter spectrogram Erol et al. (2024). However, the spectrograms used for MIR tasks are often longer, around 30 seconds or longer (Abimbola et al. (2023); Tzanetakis & Cook (2002); Knees et al. (2015)). In addition, MoMamba removes the use of positional encoding, due to the fact that Mamba processes the spectrogram sequentially across the temporal dimension. Furthermore, MoMamba does not patchify the input spectrogram, removing another reason to utilize positional encoding. By patchifying inputs and adding unnecessary positional encoding, AudioMamba fails to exploit the strengths of SSMs. We provide ablations with these claims in Table 1. With these changes to the overall structure, we show that MoMamba outperforms AudioMamba in MIR tasks.

### 3.2.1 Classification Token

Rather than patching the spectrogram as in AudioMamba (Erol et al. (2024)), we keep the whole sequence in order to preserve temporal continuity. The features are embedded into a higher-dimensional space. Then, a class token is appended to the end of the sequence.

Classification tokens are regularly used in classification models such as BERT Devlin et al. (2019) and AudioMamba (Erol et al. (2024)), where the information is aggregated into the token. Let $c \in \mathbb{R}^D$ be the classification tokens. We append this to the input signal $u \in \mathbb{R}^{L \times D}$ to form $u' = (u_0, u_1, \dots, u_L, c)$, where $u'_i$ is the $i$-th timestep. From this, we can deduce that after one layer, the forward Mamba calculates

$$
y_{L+1} = c' = C\overline{A}^{L+1}\overline{B}u'_0 + C\overline{A}^L\overline{B}u'_1 + \cdots + C\overline{B}c
\tag{3}
$$

and the backwards Mamba calculates

$$c' = C\overline{A}^{L+1}\overline{B}u'_L + C\overline{A}^L\overline{B}u'_{L-1} + \cdots + C\overline{B}c. \tag{4}$$

Note that the backwards Mamba layer maintains the classification token at the end of the sequence. From both layers, we now have two sequences: $y_{\text{forward}} \in \mathbb{R}^{L+1 \times D}$ and $y_{\text{backward}} \in \mathbb{R}^{L+1 \times D}$. To combine these into one sequence, we concatenate on the channel dimension and then project back to the original input shape. The final equation for one passthrough of a bidirectional block is

$$y_{\text{bidirectional}} = \text{Linear}(\text{Concat}((y_{\text{forward}}, y_{\text{backward}}))). \tag{5}$$

After passing through all the bidirectional Mamba layers, we obtain $y_{\text{output}} \in \mathbb{R}^{L+1 \times D}$. Then, we select only the classification token index, which contains the accumulation of the information from the input sequence. The resulting classification token vector is then projected to the number of classes.

### 3.2.2 MAMBA (SSM) LAYERS

After the embedding, the transformed data is passed through a stack of Bidirectional Mamba blocks, shown in the middle group in Figure 1. Each block leverages state-space modeling to capture both short- and long-term dependencies (Gu & Dao (2024)). Following each block, Layer Normalization (Ba et al. (2016)) is applied to stabilize the input distribution for the subsequent block, and a Dropout layer (Hinton et al. (2012)) is used for regularization. This block structure is repeated $N$ times, where $N$ is the number of Mamba blocks, thereby increasing the model's depth and capacity to learn complex temporal patterns.

### 3.2.3 BIDIRECTIONAL MAMBA BLOCK

In each Bidirectional Mamba block, the model processes the sequential data in both forward and backward directions. The classification token remains at the end of the sequence, allowing it to aggregate information from both directions. The outputs from the forward and backward passes are concatenated and passed through a linear layer to project back to the input shape.

### 3.2.4 CLASSIFICATION/REGRESSION HEAD

After passing through the Mamba layers, we extract the final timestep of the output sequence to get the classification token and project it to match the class size using a fully connected layer. This step ensures that the extracted representations are mapped to the appropriate number of output classes, allowing the model to make accurate predictions.

For regression, we use a two-layer feedforward head with ReLU and dropout, projecting the embedding to a single scalar via a sigmoid.

## 4 EXPERIMENTS

### 4.1 DATA PROCESSING

For each input audio sample, we generate a mel-spectrogram using *Librosa*'s (McFee et al. (2025)) methods to capture its frequency content over time. We generate the log-mel spectrograms with 128 mel bands using a 60 ms Hann window and a 10 ms hop size at a sampling rate of 44.1 kHz, matching the spectrogram parameters of (Huang et al. (2022)). The STFT was computed with an FFT size equal to the window length, and the resulting mel spectrograms were converted to the log scale (dB) with the maximum power as the reference.

### 4.2 MODEL SETUP AND PARAMETERS

To prepare for training, we tune the learning rate, the embedding dimension size, and the number of bidirectional Mamba blocks. The embedding dimensions and the number of bidirectional Mamba blocks we searched through are $256, 512, 1024$ and $3, 4, 5, 6$, respectively. We find that the number of parameters is generally around 40 million for the best setup in each task.

### 4.3 TRAINING AND TESTING

For each downstream task, we train each model from scratch. We also use data augmentation techniques to increase the number of samples in our training set. For each task, the specifics of the techniques will be described in the following paragraphs. For classification tasks, we use cross-entropy loss, and for regression tasks, we use mean squared error loss. For all models, we use AdamW (Loshchilov & Hutter (2019)) as our optimizer. To maintain data integrity, we carefully ensure that all augmented versions of a sample remain within the same split, preventing any potential data leakage and preserving the validity of our evaluations.

### 4.4 BASELINE METHODS

We select multiple baseline models for comparisons. MusicResNet (Elbir & Aydin (2020) )is a CNN-based model that was evaluated on GTZAN. However, they do not use the fault-filtered split resulting in high scores. DenseNet (Bian et al. (2019)) has been shown to perform well on MIR tasks. The CRNN (Choi et al. (2017)) combines CNNs and RNNs, resulting in promising performance. We also compare to LSTM (Hochreiter & Schmidhuber (1997)) as an additional baseline. AST (Gong et al. (2021)) has been shown to utilize transformers effectively for audio tasks. AudioMamba (Erol et al. (2024)) has shown that Mamba can be applied to speech and sound effects tasks. We train and evaluate each of these models from scratch on all of the following MIR problems. We also hyperparameter-tune the models to evaluate their performance under our setup.

### 4.5 DATASETS AND DOWNSTREAM TASKS

To evaluate these models, we use seven different downstream tasks: meter classification, genre classification, global key classification, emotion regression for valence and arousal, instrument classification, and pitch classification. We use the following commonly used datasets:

**Meter2800** (MTR2800)
We use the Meter2800 dataset (Abimbola et al. (2023)), which includes 1200 samples each of 3-meter and 4-meter, and 200 samples each of 5-meter and 7-meter. Every sample is 30 seconds long. The samples are taken from various raw audio datasets, one example being GTZAN (Tzanetakis & Cook (2002)). We pitch shift each sample in the training set by $[-1, 0, 1]$ semitones and time-stretch by a $[0.75, 1, 1.25]$ multiplier, resulting in 9 augmented samples for each original sample.

**GTZAN**
We use the GTZAN dataset (Tzanetakis & Cook (2002)), a widely used benchmark for music genre classification. However, we are aware of the well-documented issues with this dataset, including mislabeled samples, duplicates, and potential distortions (Sturm (2014)). To mitigate these concerns and ensure a more reliable evaluation, we adopt the train-test split proposed by Kereliuk et al. (Kereliuk et al. (2015)), which aims to address some of these limitations. The dataset consists of 1000 30-second audio samples, evenly distributed across 10 distinct genre classes, making it a balanced dataset for classification tasks.

**GiantSteps**
We use the GiantSteps dataset, containing 604 samples, (Knees et al. (2015)) for testing and the GiantSteps-MTG (Korzeniowski & Widmer (2017)) dataset, containing 1486 samples, for training to follow the same setup as (Korzeniowski & Widmer (2017)) for global key classification. We augment the training set by pitch-shifting each sample to the other 11 keys, improving the model's ability to generalize for this downstream task. When evaluating this task, we use the weighted score proposed by Raffel et al. (2014), which takes into account the relationship between the predicted and true key and mode of a given sample.

**EmoMusic**
For emotion regression tasks, we use the EmoMusic Dataset (Soleymani et al. (2013)), which contains the mean valence and arousal score for 744 45-second samples. We use the given train-test split. For this task, we concatenate the chromagram to the input spectrogram. Without concatenating the chromagram to the spectrogram, we found that many models fail to capture the harmonic content necessary for learning meaningful valence representations. Additionally, we normalize the labels to be in the range $[0, 1]$.

**NSynth**

We use the NSynth dataset (Engel et al. (2017)), (305,979 samples), consisting of multiple pitches and instruments. It also contains synthetic and acoustic versions of each instrument. Each sample is monophonic and is 4 seconds in length. We did not use any data augmentation for this dataset. We use the given train-test splits to allow direct comparison to existing benchmarks.

## 4.6 ABLATIONS ON MODEL DESIGN

We conduct a series of ablations to study the impact of different architectural design choices. Specifically, we evaluate the model with and without a class token, vary the placement of the class token, test the inclusion and removal of positional encodings, and compare unidirectional versus bidirectional variants. These ablations allow us to isolate how each component contributes to performance and provide insights into which design elements are most critical for capturing musical structure.

| Model Setup | $\text{NSynth}_P$ (Acc.) |
| --- | --- |
| MoMamba | 0.937 |
| + Positional Encoding | 0.887 |
| - Class Token | 0.835 |
| + Class Token (front of sequence) | 0.017 |
| + Class Token (middle of sequence) | 0.906 |
| Unidirectional | 0.903 |

Table 1: Ablation study on MoMamba design choices. Performance is measured on NSynth Pitch classification (Accuracy).

Adding positional encoding to the model does not improve model performance. Bidirectionality improves the performance of Mamba by a significant margin. Finally, placing the class token at the front prevents effective aggregation of information, as reflected in the ablation study. More ablations are located in the appendix (A.1).

## 5 RESULTS

### 5.1 ANALYSIS AND EVALUATION OF MOMAMBA

The results of MoMamba and other non-pretrained models presented in Table 2, with comparisons to pretrained models reported separately in Table 3. Across all tasks, MoMamba consistently outperforms other non-pretrained models and achieves competitive performance relative to pretrained models. Considering the aggregated results, MoMamba exhibits a significantly higher average performance than all other non-pretrained models. It demonstrates particularly strong results on note-based classification tasks while maintaining comparable performance on more subjective tasks, such as emotion prediction. These findings suggest that MoMamba's architecture effectively captures both local and long-range musical dependencies without relying on large-scale pretraining, and enabling it to represent both objective structural information and higher-level perceptual attributes of music.

### 5.2 INFERENCE TIME COMPARISON OF MOMAMBA AND OTHER MODELS

We report inference time comparisons on a subset of tasks in the appendix (5). As expected, MoMamba and AudioMamba (Erol et al. (2024)) achieve significantly faster inference than AST (Gong et al. (2021)), while remaining slower than lightweight CNN-based models such as MusicResNet (Elbir & Aydin (2020)) and CRNN (GitYCC (2019)). Mamba-based models process full sequences with linear-time complexity, enabling them to capture long-range dependencies more efficiently than transformers, but at the cost of some raw speed compared to highly parallelized CNNs.

| Dataset | MTR2800 | GTZAN | GS | EMO$_V$ | EMO$_A$ | NSynth$_P$ | NSynth$_I$ | Average |
|---|---|---|---|---|---|---|---|---|
| Metric | Acc. | Acc. | W. Acc. | R2 | R2 | Acc. | Acc. | |
| MusicResNet 11 | 0.694 | 0.619 | 0.304 | NA* | NA* | 0.206 | 0.016 | — |
| DenseNet 21; 4 | 0.808 | 0.433 | 0.218 | 0.159 | 0.283 | 0.239 | 0.091 | 0.319 |
| CRNN 14; 6 | 0.741 | 0.646 | 0.138 | 0.060 | 0.370 | 0.906 | 0.734 | 0.514 |
| LSTM 20 | 0.672 | 0.366 | 0.302 | 0.120 | 0.427 | 0.745 | 0.719 | 0.479 |
| AST 15 | 0.694 | 0.617 | 0.253 | 0.228 | **0.549** | 0.884 | 0.743 | 0.567 |
| AudioMamba 13 | 0.650 | 0.567 | 0.451 | 0.064 | 0.193 | 0.897 | 0.716 | 0.505 |
| **MoMamba** | **0.827** | **0.667** | **0.556** | **0.355** | 0.538 | **0.937** | **0.774** | **0.665** |

Table 2: Accuracy metrics across multiple tasks for non-pretrained models: Meter Detection (MTR2800), Genre Classification (GTZAN), Key Estimation (GS), Emotion Valence (EMO$_V$), Emotion Arousal (EMO$_A$), NSynth Pitch (NSynth$_P$), and NSynth Instrument (NSynth$_I$). Bolded numbers indicate the best performance in each column. The Average column shows the mean performance across all tasks, considering only models with results available for every task. * indicates that this model fails to perform on the task.

In summary, MoMamba provides an attractive middle ground for music classification: it delivers strong performance with substantially lower inference cost than transformers, while retaining the ability to capture global structure that CNNs often miss.

### 5.3 LIMITATIONS OF MAMBA

MoMamba achieves faster inference compared to RNNs and Transformers due to how SSMs operate, which allows it to process sequences efficiently without the sequential dependencies of RNNs or the quadratic complexity of self-attention in Transformers. Unlike RNNs, which require sequential updates, Mamba computes updates in parallel, significantly reducing latency. Additionally, it avoids the expensive attention mechanism of transformers, making it more scalable for long sequences. Despite these advantages, CNNs offer better inference speed. CNNs leverage localized convolutions and highly parallelized matrix operations, making them extremely efficient for structured data like spectrograms. While Mamba offers a strong balance of speed and long-range modeling capability, it trades some inference speed for improved performance on tasks requiring deeper contextual understanding, making it a better choice when capturing global dependencies is more important than raw efficiency.

### 5.4 DISCUSSION

MoMamba's performance across tasks highlights its strength in capturing long-range dependencies, a crucial advantage over CNN-based models that rely on localized feature extraction. Unlike convolutional architectures, which struggle to represent global musical structures, Mamba effectively learns key transitions and harmonic relationships over extended sequences.

While transformers have shown impressive results in music modeling, they require extensive pre-training on large datasets to perform well. Training a transformer from scratch often leads to suboptimal results due to its reliance on self-attention mechanisms, which demand vast amounts of data to generalize effectively (Gong et al. (2021); Copet et al. (2023); Dhariwal et al. (2020)). In contrast, MoMamba demonstrates strong performance even without pretraining, suggesting that its sequence modeling approach is inherently more data-efficient. Moreover, its linear computational complexity makes it scalable for long musical sequences compared to the quadratic cost of self-attention in Transformers. These findings reinforce Mamba's potential for real-time music analysis and suggest that future work could further enhance its capabilities through large-scale pretraining.

We reiterate that MoMamba cannot be compared directly against large pretrained models; we are not aware of any SSM pretrained on large music datasets. Future large-scale training of a Mamba model will enable such comparisons.

| Model | MTR2800 | GTZAN | GS | EMO$_V$ | EMO$_A$ | NSynth$_p$ | NSynth$_I$ |
|---|---|---|---|---|---|---|---|
| **MoMamba** | 0.827 | 0.667 | 0.556 | 0.355 | 0.538 | 0.937 | 0.774 |
| MERT-95M$^{\text{K-means}}$ 27 | – | 0.786 | 0.650 | 0.529 | 0.699 | 0.713 | 0.746 |
| MERT-95M-public$^{\text{K-means}}$ 27 | – | 0.728 | 0.673 | 0.597 | 0.725 | 0.704 | 0.756 |
| MERT-330M$^{\text{RVQ-VAE}}$ 27 | – | 0.793 | 0.656 | 0.612 | **0.747** | **0.944** | 0.726 |
| Musicset-Sup 29 | – | **0.835** | 0.286 | 0.566 | 0.726 | 0.793 | 0.731 |
| Audioset-Sup 29 | – | 0.748 | 0.667 | 0.341 | 0.545 | 0.819 | 0.676 |
| Musicset-ULarge 29 | – | 0.735 | 0.210 | 0.577 | 0.700 | 0.892 | 0.740 |
| Audioset-ULarge 29 | – | 0.672 | 0.287 | 0.438 | 0.624 | 0.805 | 0.721 |
| Musicset-USmall 29 | – | 0.686 | 0.508 | 0.389 | 0.668 | 0.824 | 0.714 |
| Audioset-USmall 29 | – | 0.797 | 0.197 | 0.386 | 0.609 | 0.777 | 0.698 |
| SOTA Excluding Above | 0.790 1 | – | **0.743** [26] | **0.617**[9] | 0.721 [5] | – | **0.782** [36] |

Table 3: Comparison of MoMamba and related models versus state-of-the-art models across various music-related tasks. Bolded numbers represent the best in the task and underlined represent the second best.

## 5.5 COMPARISON TO AUDIOMAMBA

MoMamba achieves superior performance compared to AudioMamba (Table 2). A key distinction lies in how the two models handle input structure: MoMamba processes the spectrogram as a continuous sequence, thereby preserving the natural temporal order of the data. In contrast, AudioMamba applies patchification, which disrupts global ordering and requires positional encodings to recover sequence information. Our ablations (Table 1), as well as the original AudioMamba design (Erol et al. (2024)), show that positional encodings do not improve and may possibly be detrimental to the performance of Mamba-based models on MIR tasks. Moreover, MoMamba has half the number of parameters as AudioMamba while acheiving superior performance. These results suggest that maintaining the full spectrogram structure is more effective than patch-based representations for MIR tasks.

## 5.6 COMPARISONS TO PRE-TRAINED MODELS

MoMamba demonstrates strong performance across multiple music-related tasks, notably outperforming the 95 million parameter MERT models on most benchmarks. This highlights its efficiency despite having fewer parameters than some larger pretrained alternatives. MoMamba is competitive to the pretrained models from McCallum et al. (2022), and surprisingly outperforms every Musicset and Audioset model on the NSynth tasks. MoMamba also matches with the MERT (Li et al. (2024)) models on the NSynth tasks. To make a fair comparison, we would compare a pretrained Mamba-based spectrogram model to the existing pretrained benchmarks, however, this is out of the scope of this study and left for future work. Overall, MoMamba proves to be an efficient model that delivers competitive performance despite training on small amounts of data.

## 6 CONCLUSION

State space models, specifically the Mamba model, are highly adept at capturing temporal dynamics and complex patterns in sequential musical data, making them highly effective for downstream MIR tasks. The Mamba architecture's selective scan mechanism enhances predictive accuracy and generalization, addressing key challenges in music analysis. These results underscore its potential to serve as a versatile framework for advancing MIR research. Future work may explore pretraining Mamba models on large datasets to compare with foundational models.

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

# A  METHOD

## A.1  MODEL ABLATIONS

| Model Setup | NSynth$_P$ acc. | NSynth$_I$ acc. | EmoMusic$_V$ R2 | EmoMusic$_A$ R2 |
|---|---|---|---|---|
| MoMamba | 0.937 | 0.774 | 0.355 | 0.538 |
| MoMamba (w. Unidirection) | 0.903 | 0.660 | 0.192 | 0.479 |
| + Positional Encoding | 0.887 | 0.755 | 0.270 | 0.477 |
| - Class Token | 0.835 | 0.738 | 0.036 | 0.481 |
| + Class Token (front of sequence) | 0.017 | 0.209 | 0.036 | 0.009 |
| + Class Token (middle of sequence) | 0.906 | 0.740 | 0.249 | 0.428 |

Table 4: Accuracy of different model setups on MIR tasks.

In addition to the ablations in section 4.6, we also tested the model setups on the NSynth instrument and EmoMusic tasks to further solidify our claims. The consistency of the results further show the robustness of the model setup. From this, we see that bidirectional Mamba with a class token at the end of the sequence performs the best out of every setup.

The design of AudioMamba involves placing the classification token in the middle of the input patchified spectrogram. The problem of not incorporating the entire spectrogram is addressed (13) when placing the classification token in the middle of the sequence. The bidirectionality ensures that the whole sequence is "seen" by the classification token, the first half by the forward direction and the second half by the backward direction. However, with MoMamba, we find that placing the classification token at the end of the sequence results in better performance. This may be because AudioMamba uses learnable positional embeddings, which, when the spectrogram is patchified, could misrepresent the temporal order and limit the information the classification token receives.

# B  EXPERIMENT DETAILS

## B.1  INFERENCE TIME

| Model | MTR2800 | GTZAN | GS |
|---|---|---|---|
| MusicResNet (11) | **0.73 ± 6.41** | **0.59 ± 9.1** | **2.69 ± 18.28** |
| DenseNet (21; 4) | 11.23 ± 8.68 | 5.18 ± 5.12 | 7.84 ± 23.42 |
| CRNN (14; 6) | 1.79 ± 5.56 | 1.39 ± 4.16 | 3.85 ± 19.43 |
| LSTM (20) | 72.91 ± 1.99 | 70.64 ± 7.71 | 16.72 ± 6.70 |
| AST (15) | 495.67 ± 25.88 | 64.98 ± 3.60 | 6.44 ± 17.95 |
| AudioMamba (13) | 10.34 ± 7.18 | 15.58 ± 8.70 | 5.01 ± 24.12 |
| **MoMamba** | 6.11 ± 6.92 | 9.91 ± 2.45 | 3.29 ± 12.11 |

Table 5: Average inference times (ms) with standard deviations for three tasks: Meter Detection (MTR2800), Genre Classification (GTZAN), and Key Estimation (GS). Bold numbers indicate the fastest inference in each column. Each model's inference time was recorded using a batch size of 32 and same data processing to avoid any confounding factors. Each model was evaluated on a 48 GB L40 GPU.

## B.2  DATA VISUALIZATION FOR EMOMUSIC

Below are the visualizations for both valence and arousal regression tasks for each non-pretrained model on the test set. The R2 scores for each visualization are the values in table 2 Note that AST does not learn to predict valence well even if it's score is not the lowest, as it learns to predict the average score every time. The red line indicates perfect predictions.

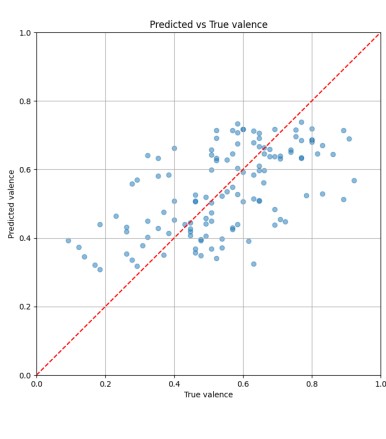

Figure 2: MoMamba Valence

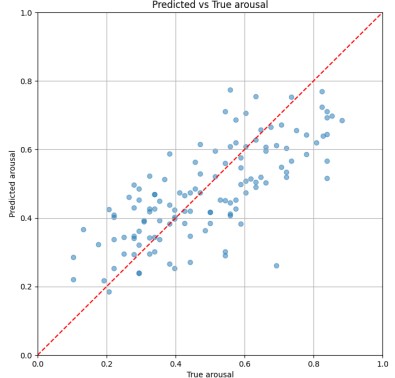

Figure 3: MoMamba Arousal

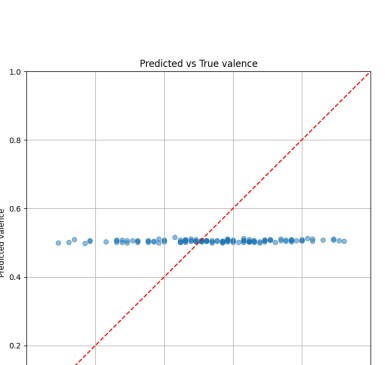

Figure 4: AST Valence

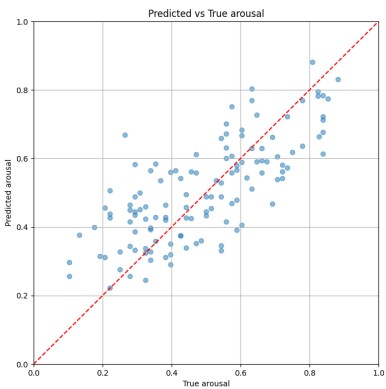

Figure 5: AST Arousal

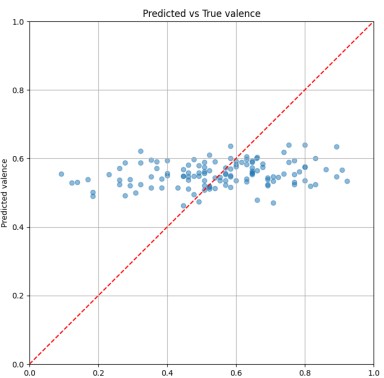

Figure 6: CRNN Valence

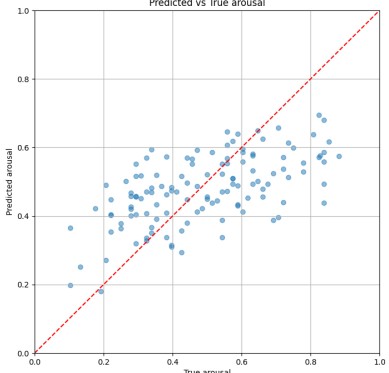

Figure 7: CRNN Arousal

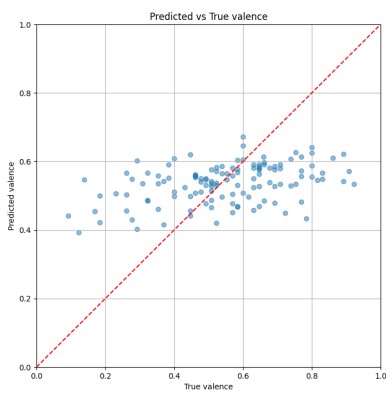

Figure 8: Densenet Valence

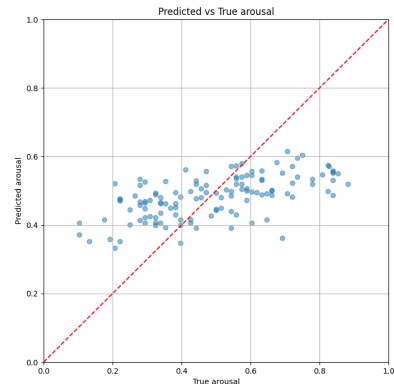

Figure 9: Densenet Arousal

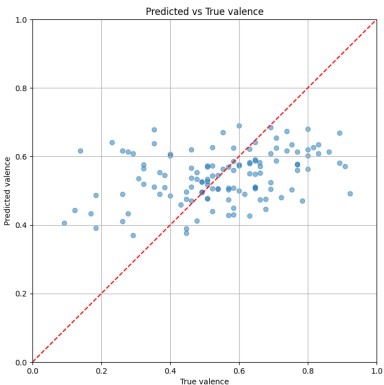
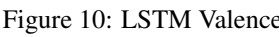

Figure 10: LSTM Valence

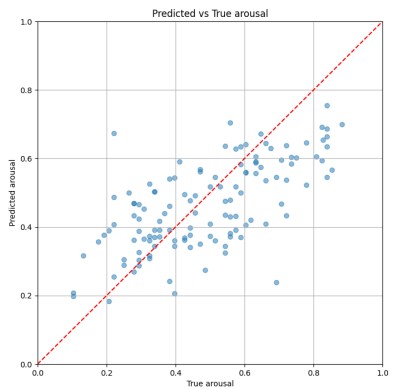

Figure 11: LSTM Arousal

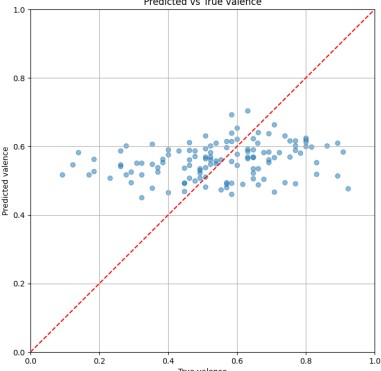

Figure 12: AudioMamba Valence

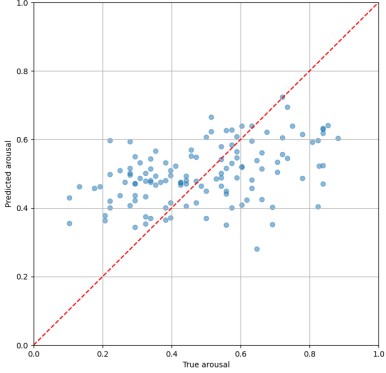

Figure 13: AudioMamba Arousal

## C  ETHICS AND REPRODUCIBILITY

### C.1  ETHICS

We do not generate any data that infringes on copyrighted material, or replacing creative work from human sources. We believe that our work can contribute positively to both researchers and users of MIR technology.

### C.2  REPRODUCIBILITY

All datasets used are publicly available. We also include a copy of both the regression and the classification MoMamba model.

### C.3  USE OF LLMS

We used LLMs to assist with creating the tables as well as fix minor grammar issues. LLMs were not used in model architecture design and training.

