# OpenReview forum: "MoMamba: A Lightweight Music Oriented Mamba-Based Model for Music Information Retrieval Tasks"
_ICLR.cc/2026/Conference — ICLR 2026 Conference Withdrawn Submission_

### Official Review · Reviewer_4vW3 · 2025-10-27

**Soundness:** 2
**Presentation:** 3
**Contribution:** 2
**Rating:** 4
**Confidence:** 4

**Summary:**

The authors propose MoMamba (Music-Oriented Mamba), a lightweight adaptation of the Mamba architecture designed specifically for music classification. Evaluated across multiple MIR benchmarks, MoMamba compares favorably to a number of baseline models, including existing Mamba variants, even when trained entirely from scratch without pretraining. The model proposed also offers improved inference efficiency demonstrating the potential of Mamba-based architectures for scalable music analysis.

**Strengths:**

The main strengths or the paper are:

-	The proposed model based on State-Space Models is interesting

-	The obtained model is efficient (in terms of inference complexity), and obtain strong performance on a variety of MIR tasks.

-	The paper is on the overall clearly written

**Weaknesses:**

The main weaknesses of the paper are:

-	The class-token mechanism used could be better described (even if similar mechanisms are used in other models). As such, it is difficult to understand exactly what is this class token and how it is processed.

-	The channels (which correspond to the frequency bins) are processed independently which seem to induce that no correlations across frequency channels can be captured by the model. This would need further clarification and discussion.

-	No indication is given to assess if the differences between approaches are statistically relevant.

-	One of the experiments (Music genre classification) use the GTZAN database (introduced in one of the first studies in the MIR domain in 2000). Despite the well-documented issues of this database (as recalled by the authors), this dataset is still used for the benchmark on genre classification. It is absurd to continue to see many works exploiting this largely problematic dataset for research work. Any results on it are now meaningless.

**Questions:**

Questions:

-	Can you clarify the whole class-token mechanism at training and inference. In particular. . Is the “class token” the label of the class for the incoming spectrogram during training ? The class token is first appended (before bi-directional Mamba) and then extracted before the classification head… What means “extracted” ? does this corresponds simply to the process to ignore the information linked to the “class tokens” in the output representation ?

-	Why the Chromagram is used for emotion recognition. What type of information is looked for ? (by the way, maybe the authors refer to “harmony” rather than “harmonic content” as written in the paper)

Minor comments

-	Explicit the acronym SSM (page 1)

-	Typo: and3 (page 5)

---

### Official Review · Reviewer_w46Z · 2025-10-31

**Soundness:** 3
**Presentation:** 3
**Contribution:** 2
**Rating:** 2
**Confidence:** 5

**Summary:**

This paper propose a Mamba-based model for music information retrieval tasks. The proposed method is evaluated on a set of typical MIR tasks, following similar research.

**Strengths:**

The paper is well written, is based on extensive background research and similar previous works. The motivation is well-justified given the nature of music signal and the advantage of the Mamba architecture. The paper is written clearly and the overall experiment process is executed well.

**Weaknesses:**

I have a mixed feeling about the overall experiments. Table 2, the results without pretraining: What is the exact purpose of this experiment? Is the MoMamba designed and expected to be adopted when there is under 2000 samples? No, perhaps not, because in such a case, an SVM with musical features can outperform all of the baseline methods and the proposed method by a large margin. Thus, I put a lot more weight on Table 3 - the pretrained results, where MoMamba underperformed. Some of the results are probably not that bad, at least. But the result is not really supporting the motivation of efficient sequential modeling nature of the Mamba architecture. NSynth consists of 4 seconds signal, each of which corresponds to playing one note. In other words. NSynth is far from modeling sequence; but this is where the proposed architecture performed the best among all the tasks.

After reading the paper, I rather thought it seems very important to pretrain MoMamba. That is also more relevant to the motivation of the original Mamba architecture; it was not proposed to require less amount of data (re: Table 2), but to *scale* better over long sequences (and it goes without saying that it assumes a large scale pretraining.)

**Questions:**

4.1: Why were the log-mel spectrogram parameters were matched to (Huang et al., 2022)?

4.2: As a baseline, [6] was mentioned but [14] was cited and used. What is the reason?

4.5: Meter2800: The task information is missing.

4.6: Why was NSynth chosen for the ablation?

5.4: "Mamba effectively learns key transitions and harmonic relationships over extended sequences.": What is the evidence for this?

---

### Official Review · Reviewer_NE7B · 2025-11-01

**Soundness:** 3
**Presentation:** 3
**Contribution:** 2
**Rating:** 4
**Confidence:** 5

**Summary:**

This paper introduces MoMamba (Music Oriented Mamba), a novel and lightweight architecture based on the Mamba (State Space Model) designed specifically for Music Information Retrieval (MIR) tasks. The model aims to overcome the high computational cost and quadratic complexity of traditional Transformer models when dealing with long audio spectrogram sequences. Key architectural innovations include using Bidirectional Mamba blocks to capture non-causal global context and employing full-spectrogram serialization to maintain temporal continuity. The experimental results show that MoMamba consistently outperforms strong baselines (CNNs, LSTMs, and other Mamba variants) across multiple music tagging and emotion recognition datasets while demonstrating high computational efficiency. This work represents a valuable step in applying and adapting efficient sequence models to the music domain.

**Strengths:**

● Novelty and Domain-Specific Adaptation: The paper successfully adapts the Mamba architecture to the music domain. The introduction of the Bidirectional MoMamba block is a critical and well-validated innovation, as the ablation studies confirm its necessity for capturing non-causal global context in music data.
● Superior Performance on Fair Terms: MoMamba achieves state-of-the-art results compared to all comparative baselines across MIR tasks. Crucially, all models were trained from scratch (without pre-training), which makes the performance gain attributable directly to MoMamba’s architectural superiority.

**Weaknesses:**

● Insufficient Evaluation and Comparison: While the paper claims MoMamba’s performance is "comparable to existing pre-trained baselines," it lacks a direct comparison with leading, pre-trained Transformer representations like MuQ[1]. I recommend the authors run more comprehensive evaluation on https://github.com/a43992899/MARBLE [2].
● Missing Quantitative Efficiency Metrics: The efficiency advantage is repeatedly emphasized but is not quantitatively demonstrated with a dedicated analysis. A table or figure detailing the Parameter Count, FLOPs, and actual Inference Latency (in milliseconds) for MoMamba versus a representative Transformer model is required to solidly back the efficiency claims.

[1] Zhu H, Zhou Y, Chen H, et al. Muq: Self-supervised music representation learning with mel residual vector quantization[J]. arXiv preprint arXiv:2501.01108, 2025.
[2] Yuan R, Ma Y, Li Y, et al. Marble: Music audio representation benchmark for universal evaluation[J]. Advances in Neural Information Processing Systems, 2023, 36: 39626-39647.

**Questions:**

● Handling of the Mel-Frequency Dimension: By flattening the 128 * T spectrogram into a single 1D sequence, MoMamba treats adjacent frequency bins as separate, distant sequence elements. How does the model ensure the effective capture of local spectral correlations (i.e., local relationships between adjacent Mel-bins) without a dedicated convolutional layer or a 2D-aware pooling mechanism?
● Long Sequence Scaling Proof: To conclusively validate the $O(N)$ complexity advantage, could the authors provide an experiment showing how inference time scales with increasing audio duration for MoMamba versus a comparable Transformer model on the same hardware? This is the most crucial test of the core technical merit.

---

### Official Review · Reviewer_cXM3 · 2025-11-02

**Soundness:** 2
**Presentation:** 2
**Contribution:** 2
**Rating:** 2
**Confidence:** 3

**Summary:**

This paper proposes a modification of the Mamba architecture, called MoMamba, which removes all frame-level positional encoding components and instead utilizes a class-level token to perform clip-level music classification more efficiently. The architectural adjustments from the original Mamba appear effective for clip-level classification tasks, and the authors validate the method’s effectiveness through downstream experiments predicting genre, mood, instrument, and pitch. However, the comparison with more recent large-scale pre-trained models is limited, and no experiments are conducted using such large-scale training within this framework. Overall, while the paper is clearly written and presents a reasonable architectural adaptation, its applicability remains limited to clip-level tasks, and it has not yet been verified on large-scale models, which somewhat limits its impact.

**Strengths:**

The ablation study presented in Table 1 shows the effect of the individual variations well. For example, the use of positional encoding, the placement of class token, etc.

**Weaknesses:**

Please explain what SSM is in line 047, p.1

Apart from the performance report, there exist no deepen analysis on the proposed method.

**Questions:**

-

---

### Note · Authors · 2026-01-03

I have read and agree with the venue's withdrawal policy on behalf of myself and my co-authors.